# Balancing Generalization and Robustness in Adversarial Training via Steering through Clean and Adversarial Gradient Directions

Haoyu Tong
State Key Laboratory of Integrated
Service Networks (ISN)
Xidian University
Xi'an, China
haoyutong@stu.xidian.edu.cn

Xiaoyu Zhang*
State Key Laboratory of Integrated
Service Networks (ISN)
Xidian University
Guangdong Provincial Key
Laboratory of Novel Security
Intelligence Technologies
Xi'an, China
xiaoyuzhang@xidian.edu.cn

Yulin Jin
State Key Laboratory of Integrated
Service Networks (ISN)
Xidian University
Xi'an, China
jyl990903@163.com

Jian Lou
State Key Laboratory of Integrated
Service Networks (ISN)
Xidian University
Xi'an, China
jian.lou@hoiying.net

Kai Wu
School of Artificial Intelligence,
Xidian University
Xi'an, China
kwu@xidian.edu.cn

Xiaofeng Chen
State Key Laboratory of Integrated
Service Networks (ISN)
Xidian University
Xi'an, China
xfchen@xidian.edu.cn

## Abstract

Adversarial training (AT) is a fundamental method to enhance the robustness of Deep Neural Networks (DNNs) against adversarial examples. While AT achieves improved robustness on adversarial examples, it often leads to reduced accuracy on clean examples. Considerable effort has been devoted to handling the trade-off from the perspective of *input space*. However, we demonstrate that the trade-off can also be illustrated from the perspective of the *gradient space*. In this paper, we propose Adversarial Training with Adaptive Gradient Reconstruction (*AGR*), a novel approach that balances generalization (accuracy on clean examples) and robustness (accuracy on adversarial examples) in adversarial training via steering through clean and adversarial gradient directions. We first introduce an ingenious technique named Gradient Orthogonal Projection in the case of negative correlation gradients to adjust the adversarial gradient direction to reduce the degradation of generalization. Then we present a gradient interpolation scheme in the case of positive correlation gradients for efficiently increasing the generalization without compromising the robustness of the final obtained. Rigorous theoretical analysis proves that our *AGR* has lower generalization error upper bounds indicating its effectiveness. Comprehensive experiments empirically demonstrate that *AGR* achieves excellent capability of balancing generalization and robustness, and is compatible with various adversarial training

methods to achieve superior performance. Our codes are available at: https://github.com/RUIYUN-ML/AGR.

## CCS Concepts

• **Computing methodologies** → **Computer vision**; • **Security and privacy** → *Human and societal aspects of security and privacy*.

## Keywords

Adversarial Training, Deep Neural Networks

**ACM Reference Format:**
Haoyu Tong, Xiaoyu Zhang, Yulin Jin, Jian Lou, Kai Wu, and Xiaofeng Chen. 2024. Balancing Generalization and Robustness in Adversarial Training via Steering through Clean and Adversarial Gradient Directions. In *Proceedings of the 32nd ACM International Conference on Multimedia (MM '24), October 28-November 1, 2024, Melbourne, VIC, Australia.* ACM, New York, NY, USA, 10 pages. https://doi.org/10.1145/3664647.3680963

## 1 Introduction

Deep neural networks (DNNs) have gained widespread adoption in myriad multimedia processing fields thanks to their exceptional performance, such as image recognition [16, 20], text generation [1, 2], speech recognition [34, 61] and object detection [19, 50]. However, due to widely recognized vulnerabilities caused by various attacks, the security concerns and privacy protection issues for AI-backed multimedia systems are increasingly aggravating [12, 31, 32, 60]. Currently, a wide range of real-world applications have been shown to be vulnerable to adversarial examples (AEs) [13, 18, 21, 40], which adds tiny perturbations to clean examples that are imperceptible to human perception and give a false prediction. There exist considerable well-established methods for adversarial example generation such as FGSM [13], PGD [35], C&W [6], AutoAttack [9], *etc.* The introduction of adversarial examples has spurred the development of numerous techniques to ensure the trustworthiness of multimedia processing [14, 58, 59], research on the adversarial attack defense mechanisms has been a trending topic in the multimedia

*Corresponding author.

 Haoyu Tong, Xiaoyu Zhang, Yulin Jin, Jian Lou, Kai Wu, and Xiaofeng Chen

field [15, 46], among which adversarial training [35, 45, 48, 55] has been widely accepted as an effective defense.

Although AT and its variants have notably improved model robustness, they inevitably compromise generalization performance compared to standard training methodologies and vice versa. Recently, there has been a host of research dedicated to the trade-off between generalization and robustness [30, 37–39, 42, 44], such as leveraging the redundant capacity [63], instance reweighting [56, 57], adding a regularisation term for the loss function [41, 49], or leveraging mixed data [62]. However, most of the aforementioned works utilize only the information brought by the *input space* to improve the model generalization or robustness, paying little attention to the information in the *gradient space* and even less attention to the dynamics of the *gradient space*, which is also very important in influencing the model performance. In order to thoroughly investigate the trade-off issue from the perspective of *gradient space*, we conducted an experiment with the results depicted in Figure 1. We observe that there exist various levels of correlation between the gradient direction of the clean and adversarial robust loss during the adversarial training process, and even a negative correlation between their directions in some cases, which exactly explains that the adversarial training updating process leads to the gradual rise of the robustness and the degradation of the generalization. We perceive this finding as an indication of the existence of the trade-off problem, which can be explained from the perspective of the difference in gradient directions between clean loss and adversarial robust loss. In light of this phenomenon of trade-off observed from the gradient direction, we ask the intuitive question: *"Can we exploit these gradient directions to control or improve the trade-off in the training process dynamically?"* As an answer to the question, we are tackling the trade-off problem from a novel perspective by dividing the clean and adversarial directions of the gradient into two cases during adversarial training.

- In the negative correlation case, since the reduction in generalization comes from this situation, it is possible to adjust the direction of the adversarial gradient to remove that negative effect and maintain a positive correlation with the original gradient direction.
- In the positive correlation case, correlation-based gradient interpolation can be employed to enhance generalization, paying high attention to increasing generalization and low attention to maintaining robustness in high correlation and vice versa.

Based on the above inspiration, we propose a novel adversarial training method based on adjusting the direction of the gradients called Adversarial Training with Adaptive Gradient Reconstruction (*AGR*) that aims to improve the generalization without compromising the robustness of the final obtained. Our framework divides the parameter update process into two cases based on the cosine similarity between gradient pairs (see Figure 2). **1) Negative correlation case**, we propose the Gradient Orthogonal Projection (GOP) to decompose the adversarial gradient orthogonality into two parts, one aligned with the negative direction of the clean gradient and the other orthogonal to it. This achieves the desired effect that the new gradient direction causes no degradation of generalization; **2) Positive correlation case**, we propose an adaptive gradient

interpolation scheme named Gradient Interpolation Based on Cosine Similarity (GICS), which makes use of the similarity of the gradients as the weights of the interpolation. With this interpolation scheme, the adversarial training process can dynamically pay more (or less) attention to robustness and less (or more) attention to generalization in the presence of low (or high) correlation, achieved by assigning interpolation weights corresponding to the gradient. This ensures efficient and dynamic improvement of generalization without compromising the robustness of the final obtained in adversarial training. The main contributions of our work can be summarized as follows:

- We analyze the trade-off problem that exists during adversarial training from a novel perspective, *i.e.*, *gradient space*, utilizing this perspective as a breakthrough point to further control or improve the trade-off.
- We propose an innovative adversarial training method named *AGR* to increase the generalization without compromising the robustness of the final obtained. Additionally, this approach is compatible with most of the existing adversarial training methods to achieve outstanding performance.
- We theoretically prove that our *AGR* has lower generalization error bounds compared to other AT methods. We also conducted extensive experiments to evaluate the *AGR* against five state-of-the-art adversarial attacks, demonstrating superior performance in handling the trade-off.

## 2 Related Work

**Adversarial Attacks.** Adversarial examples [40] have become a prevalent attack method imposing tiny perturbations to the model inputs that drive the target model mispredicts outputs. There has been a considerable amount of literature on the methods of their generation since their discovery. One of the earliest and most well-known methods is the Fast Gradient Sign Method (FGSM) [13]. Most subsequent adversarial attacks have been proposed based on this approach. Iterative FGSM (I-FGSM) [23] acts as an extended variant of the FGSM, which adds small perturbations by iteratively using the FGSM. PGD [35] is to randomly initialize a point on the input neighborhood as the starting point and then apply I-FGSM. Unlike FGSM [13], C&W [6] is an optimization-based attack that transforms the attack problem into a minimization problem with constraints, instead of making use of the model's gradient information to generate perturbations.

**Adversarial Defense.** In order to address and mitigate the potential threat of adversarial example attacks, manifold defense methods have been introduced and proposed. These include adversarial training [35, 45, 48, 55], adversarial detection [7, 51], certified adversarial robustness [8, 24, 26] and adversarial purification [3, 36, 52]. Among these techniques, adversarial training has emerged as a particularly effective defense method that involves training a model to improve its robustness by utilizing adversarial examples as training data. There are numerous approaches to adversarial training, each with its own unique strengths. Among them, PGD-AT [35] is the earliest and most widely accepted method, which improves model robustness by using maximization with PGD attack. TRADES [55] proposes a surrogate loss by analyzing the upper bound of the robust error, which pushes the decision boundary away from the examples. AWP [48] reveals a clear correlation between weight

landscape and robust generalization gap. By smoothing the weight loss landscape, the generalization gap can be effectively reduced to improve robustness. CFA [47] investigates the effects of adversarial perturbations on different classes, and improves the robustness by customizing the perturbation configurations of different classes during adversarial training.

## 3 Methodology

### 3.1 Preliminaries

Consider a $K$-class classification task on a dataset $\mathbb{D} = \{(x_i, y_i)\}_{i=1}^n \subseteq X \times \mathcal{Y}$, where $x_i \in \mathbb{R}^d$ represents examples drawn from a defined unknown distribution, and $\mathcal{Y}$ represents all possible labels corresponding to the examples in $X$. The prediction of input data $x$ on model $f$ is denoted as $f(w; x)$, where $w \in \mathbb{R}^p$ represents the weight of the model. For an adversarial example classification problem, we use $x'$ to denote the adversarial example of $x$.

The complete clean loss function in standard training is defined as:

$$\mathcal{L}_{std}(w) = \frac{1}{n} \sum_{i=1}^n \ell(f(w; x_i), y_i), \qquad (1)$$

where $n$ is the number of the training data, $f(w; \cdot)$ is the model (neural network), $\ell(\cdot)$ is the loss function (*e.g.*, the cross-entropy (CE) loss).

For adversarial training, we denote the adversarial (or robust) loss as:

$$\mathcal{L}_{adv}(w) = \frac{1}{n} \sum_{i=1}^n \max_{\|x'_i - x_i\|_p < \epsilon} \ell(f(w; x'_i), y_i), \qquad (2)$$

where $x'$ is considered an adversarial example that falls within the $\epsilon$-ball, bounded by $L_p$-norm and centered at the clean example $x$.

### 3.2 Overview

Adversarial training has been demonstrated to be an effective technique for enhancing a model's robustness against adversarial examples. However, one potential drawback of this approach is that it may decline the model classification accuracy on clean examples. This is because the model may become overly focused on defending against adversarial attacks, which can negatively impact its ability to correctly classify clean examples.

To address this issue, we propose a novel adversarial training approach with Adaptive Gradient Reconstruction (*AGR*). Specifically, *AGR* introduces orthogonal projection and gradient interpolation in adversarial training, which modifies the gradient of robust loss during the adversarial training process. As a result, *AGR* improves standard accuracy without compromising the robustness of the final obtained. Essentially, *AGR* allows the model to better balance its attention between defending against adversarial attacks and correctly classifying clean examples.

We first explore the application of orthogonal projection methods in adversarial training. This exploration provides a metric for distinguishing between positive and negative correlation gradient pairs. For convenience, we denote the gradient of the latter concerning $\mathcal{L}_{std}(w)$ and $\mathcal{L}_{adv}(w)$ as $\mathcal{L}_n$ and $\mathcal{L}_{adv}$, respectively.

*Definition 3.1.* The **cosine similarity** between the gradients $\nabla \mathcal{L}_n$ and $\nabla \mathcal{L}_{adv}$ is $\Psi(\nabla \mathcal{L}_n, \nabla \mathcal{L}_{adv}) = \frac{\nabla \mathcal{L}_n \cdot \nabla \mathcal{L}_{adv}}{\|\nabla \mathcal{L}_n\|_2 \|\nabla \mathcal{L}_{adv}\|_2}$.

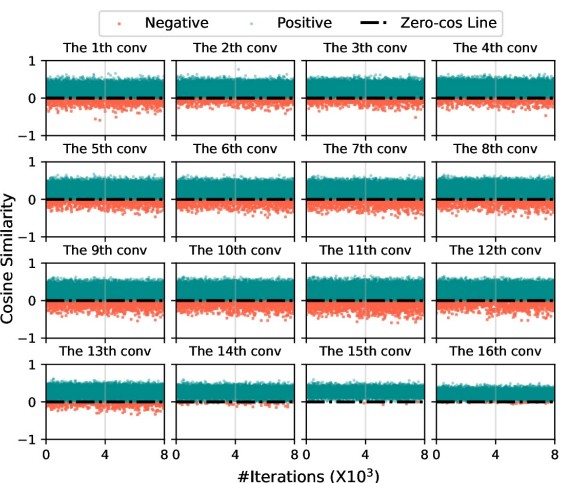

**Figure 1: The cosine similarity of $\nabla \mathcal{L}_n$ and $\nabla \mathcal{L}_{adv}$ of weights of convolutional layers of PreActResNet-18 trained on CI-FAR10 by TRADES. Below each scatter plot is the value of cosine similarity in the corresponding iteration rounds.**

When the $\Psi(\cdot) < 0$, it indicates that the gradient is in the opposite direction for standard and adversarial training. We refer to this gradient pair as a negative correlation gradient pair. Whereas, when $\Psi(\cdot) > 0$, we call the gradient pair a positive correlation gradient pair. We also conduct a simple experiment in Figure 1 to show that there are many cases where the $\Psi(\cdot) < 0$ during the model training process.

In the adversarial training process, updating the model in the direction of $\nabla \mathcal{L}_{adv}$ will significantly affect its predictive performance at point $x'$, but will result in a smaller change to the prediction of $x'$ along the orthogonal to $\nabla \mathcal{L}_{adv}$. We consider the effect of both directions $\nabla \mathcal{L}_{adv}$ and $\nabla \mathcal{L}_n$ on the performance of the model simultaneously. Let us use cosine similarity as a measure between gradient pairs (i.e., $\Psi(\nabla \mathcal{L}_n, \nabla \mathcal{L}_{adv})$), then we can divide the gradient pairs into two types, positive correlation gradient pairs or negative correlation gradient pairs, corresponding to $\Psi(\nabla \mathcal{L}_n, \nabla \mathcal{L}_{adv}) \geq 0$ or $\Psi(\nabla \mathcal{L}_n, \nabla \mathcal{L}_{adv}) < 0$, respectively. Subsequently, we propose the gradient orthogonal projection and gradient interpolation methods to improve the generalization for the two cases (See Figure 2). To consider the orthogonalization process in more detail, we will orthogonalize the gradient for each parameter, we have:

$$\nabla \mathcal{L}_{adv} = [\nabla \mathcal{L}_{adv}^{(1)}, \nabla \mathcal{L}_{adv}^{(2)}, ..., \nabla \mathcal{L}_{adv}^{(c)}], \qquad (3)$$

$$\nabla \mathcal{L}_n = [\nabla \mathcal{L}_n^{(1)}, \nabla \mathcal{L}_n^{(2)}, ..., \nabla \mathcal{L}_n^{(c)}], \qquad (4)$$

where $c$ is the number of the parameters.

### 3.3 Gradient Orthogonal Projection

As depicted in Figure 1, we demonstrate that there exists a few instances where the $\Psi(\nabla \mathcal{L}_n^{(i)}, \nabla \mathcal{L}_{adv}^{(i)}) < 0$ during adversarial training. It suggests that this update moving will increase the robustness but decrease the generalization. To reduce the loss of generalization, we propose to use "orthogonalize" to refine the gradient $\nabla \mathcal{L}_{adv}^{(i)}$.

We now introduce the **G**radient **O**rthogonal **P**rojection (GOP) clearly and concisely, using a formulaic approach. We first orthogonally project $\nabla \mathcal{L}_{adv}^{(i)}$ along $\nabla \mathcal{L}_n^{(i)}$ to obtain the new direction $g_1^i$

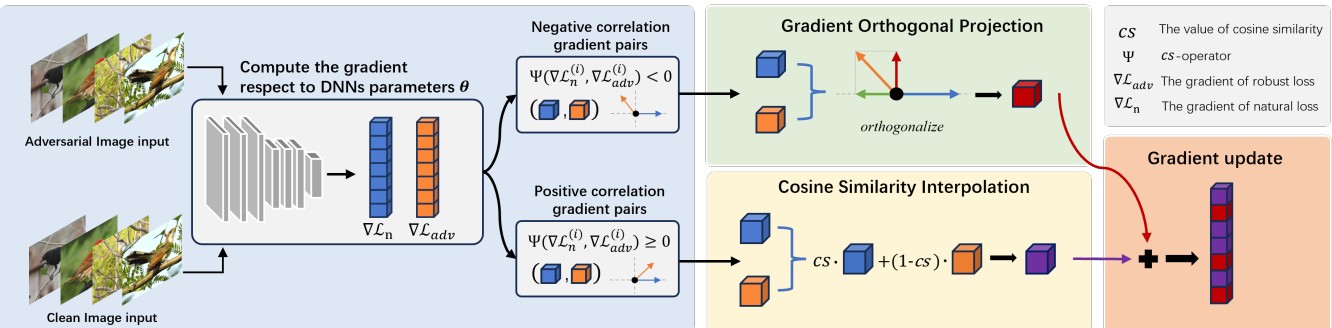

**Figure 2: Overview of the proposed *AGR*.**

as follows:

$$g_1^i = \frac{\langle \nabla \mathcal{L}_{adv}^{(i)}, \nabla \mathcal{L}_n^{(i)} \rangle}{\langle \nabla \mathcal{L}_n^{(i)}, \nabla \mathcal{L}_n^{(i)} \rangle} \nabla \mathcal{L}_n^{(i)}. \qquad (5)$$

In this scenario, the vectors of $g_1^i$ and $\nabla \mathcal{L}_n^{(i)}$ are exactly opposite in direction, leading to reduced predictive accuracy on the model's clean examples. So we discard the component $g_1^i$ and choose the component $\nabla \mathcal{L}_{adv}^{(i)}$ that orthogonal to $\nabla \mathcal{L}_n^{(i)}$ as the moving direction, we denote $Proj(\cdot)$ as follows:

$$g_2^i = \nabla \mathcal{L}_{adv}^{(i)} - g_1^i. \qquad (6)$$

It is worth noting that we take $g_2^i$ as the gradient of the *i*-th parameter update for this model does not cause much decrease in robustness as it is positively correlated with $\nabla \mathcal{L}_{adv}^{(i)}$ (*i.e.*, $\Psi(g_2^i, \nabla \mathcal{L}_{adv}) > 0$).

In the case where $\Psi(\nabla \mathcal{L}_n^{(i)}, \nabla \mathcal{L}_{adv}^{(i)}) > 0$, let's reconsider using orthogonal decomposition for $\nabla \mathcal{L}_{adv}^{(i)}$ with respect to $\nabla \mathcal{L}_n^{(i)}$. The direction of $g_1^i$ aligns with $\nabla \mathcal{L}_n^{(i)}$, indicating it doesn't hinder the model's prediction on clean examples. Since $g_2^i$ is orthogonal to $\nabla \mathcal{L}_n^{(i)}$, it has minimal impact on clean prediction performance but positively correlates with $\nabla \mathcal{L}_{adv}^{(i)}$, enhancing robustness. Thus, discarding any component via gradient orthogonal projection negatively affects generalization or robustness (e.g., discarding $g_1^i$ reduces generalization). Therefore, applying gradient orthogonal projection when $\Psi(\nabla \mathcal{L}_n^{(i)}, \nabla \mathcal{L}_{adv}^{(i)}) > 0$ is inappropriate, and we next consider a gradient interpolation method for this scenario.

While there exists a series of works that have used the GOP to solve a variety of problems. For example, Bryniarski *et al.* [5] explores a new attack technique aimed at constructing adversarial examples that satisfy multiple constraints simultaneously through GOP. Farajtabar *et al.* [11] proposes to use it to solve the *catastrophic forgetting* in continual learning. Li *et al.* [27] introduces subspace learning to federated unlearning via the orthogonal projection. To the best of our knowledge, we are the first work to pursue the improvement of generalization through gradient orthogonal projection.

### 3.4 Gradient Interpolation

Let's focus on improving generalization during training. After performing gradient orthogonal projection in adversarial training, the model achieves higher standard accuracy and consistent robustness compared to AT. However, it is worth noting that the increase in

standard accuracy is often not very significant. Figure 1 shows that during the training process, only a small fraction of iterations have $\Psi(\cdot)$ values less than 0.

Overall, there are still a large number of positive correlations in the direction of the gradient here, and we will make reasonable utilization of this gradient information to further handle the trade-off. Our main idea is to consider gradient interpolation based on cosine similarity between $\nabla \mathcal{L}_{adv}^{(i)}$ and $\nabla \mathcal{L}_n^{(i)}$. Through interpolation, we can obtain a gradient that is very correlated with both gradient vectors, which can effectively balance the robustness and generalization. Thus, we propose the **G**radient **I**nterpolation Based on **C**osine **S**imilarity (GICS), the interpolation of the formula can be written as,

$$G = cs \cdot \nabla \mathcal{L}_n^{(i)} + (1 - cs) \cdot \nabla \mathcal{L}_{adv}^{(i)}, \qquad (7)$$

where *cs* is the cosine similarity between $\nabla \mathcal{L}_{adv}^{(i)}$ and $\nabla \mathcal{L}_n^{(i)}$.

The above interpolation method is able to dynamically handle the balance between generalization and robustness by using the cosine similarity between $\nabla \mathcal{L}_{adv}^{(i)}$ and $\nabla \mathcal{L}_n^{(i)}$ as interpolation weighting factors. When the cosine similarity between gradients is larger, due to the larger *cs* this interpolation method will consider $\nabla \mathcal{L}_n^{(i)}$ more and efficiently gain generalization without loss of robustness. When the cosine similarity between gradients becomes small, the interpolation method will give priority to $\nabla \mathcal{L}_{adv}^{(i)}$ to prevent the decrease in robustness. Thus, it achieves the effect of improving generalization without loss of robustness. And when $\Psi(\nabla \mathcal{L}_n^{(i)}, \nabla \mathcal{L}_{adv}^{(i)})$ is less than 0, we do not opt for using gradient interpolation. We reason that the gradient pairs are negatively correlated at this moment, and the interpolated gradient will retain a small similarity to the clean gradients or adversarial robust gradients. This similarity makes it challenging to improve the generalization and robustness of the model.

No matter which interpolation method, it will lead to a certain difference between the new gradient direction and the original $\nabla \mathcal{L}_{adv}^{(i)}$, which will cause the model to fail to obtain the robustness in time, so we choose to take GICS after the model obtains the maximum robustness (*e.g.*, in the subsequent experiments we start using GICS at 150 epochs) and in order to finally maintain higher robustness, we choose to perform a gradient clipping on the interpolated gradient in $l_2$ norm, *i.e.*, the *Clip* operation is formulated as $G/max(1, \frac{\|G\|_2}{C})$, for a clipping threshold $C$. It is well documented that catastrophic overfitting of AT is related to the magnitude of the gradient paradigm, and reducing the magnitude

of the gradient paradigm can effectively suppress catastrophic forgetting [29]. To better facilitate understanding, the pseudo-code for *AGR* is presented in Supplementary.

## 4 Theoretical Analysis

In this section, we provide a general result (Theorem 4.1) to demonstrate the effectiveness of *AGR* in improving generalization while maintaining robustness during each iteration. Then, we use Rademacher complexity to bound the generalization error of *AGR*. Now, we consider generalization in terms of two parts of *AGR* to obtain the following results:

**Theorem 4.1.** *Let $\nabla \mathcal{L}_n, \nabla \mathcal{L}_{adv}$ denote the clean and adversarial gradients, respectively. For arbitrary cases of $-1 \leq \Psi(\nabla \mathcal{L}_n^{(i)}, \nabla \mathcal{L}_{adv}^{(i)}) < 0$ and $0 \leq \Psi(\nabla \mathcal{L}_n^{(i)}, \nabla \mathcal{L}_{adv}^{(i)}) \leq 1$, AGR with gradient $G$ for one iteration, it holds that $G$ induces a descent in both $\mathcal{L}_n$ and $\mathcal{L}_{adv}$.*

Theorem 4.1 shows that both in gradient orthogonal projection and cosine similarity interpolation operations, it is at least guaranteed that the direction of the gradient update causes no increase in clean loss and adversarial robust loss. Detailed notations and proof can be found in Supplementary. Next, we consider the following setting of Rademacher complexity.

There has an unknown distribution $\mathcal{D}$ over the input space $\mathcal{Z}$, from which we drawn $N$ examples i.i.d from $\mathcal{D}$ to form the standard training dataset $S = \{z_1, z_2, ..., z_N\}$ where $z_i = (x_i, y_i)$. Similarly, we denote the adversarial training dataset $S'$ drawn from the distribution $\mathcal{T}$. We formulate the population risk and empirical risk as:

$$R_{\mathcal{D}}(f) = E_{(x,y) \sim \mathcal{D}} [\ell(f(\theta, x), y)], \quad (8)$$

$$R_S(f) = \frac{1}{n} \sum_{i=1}^{N} \ell(f(\theta, x_i), y_i), \quad (9)$$

where $\ell(\cdot, \cdot)$ represents the loss function.

In our setting, *AGR* utilizes the standard training dataset and adversarial training dataset during adversarial training denoted as $S$ and $S'$, respectively. Further, *AGR* maintains the value of $\mathcal{L}_{std}$ without decreasing during the optimization of $\mathcal{L}_{adv}$, thus we could view $f_{AGR}$ as learned from $S$ and $S'$,

$$f_{AGR} = \underset{f \in \mathcal{F}}{\arg\min} \, R_{S+S'}(f) = \underset{f \in \mathcal{F}}{\arg\min} \, \lambda R_S(f) + (1-\lambda) R_{S'}(f), \quad (10)$$

where $\mathcal{F}$ represents the function space and $\lambda$ denotes the proportion between the standard training dataset and the adversarial training dataset. Following this, we give the definition of empirical Rademacher complexity and derive the generalization error of *AGR* based on Rademacher complexity.

*Definition 4.2.* Given a unknown distribution $\mathcal{D}$ and a function space $\mathcal{F}$, let $S = \{z_i\}_{i=1}^{N}$ denotes the training dataset drawn i.i.d from $\mathcal{D}$ and $\{\sigma_i\}_{i=1}^{N}$ be the independent random variables set drawn uniformly from $\{-1, 1\}$. Then, the empirical Rademacher complexity of $\mathcal{F}$ on the set $S$ is defined to be:

$$\hat{\mathfrak{R}}_S(\mathcal{F}) = E_{\sigma} \sup_{f \in \mathcal{F}} \left[ \frac{1}{N} \sum_{i=1}^{N} \sigma_i f(z_i) \right]. \quad (11)$$

**Theorem 4.3.** *Assume that $\mathcal{F}$ is a function space with the range [0,1], let $D^{N_s} = \{z_n^s\}_{n=1}^{N_s}$ and $D^{N_a} = \{z_n^a\}_{n=1}^{N_a}$ be two datasets of i.i.d*

*sampled from the standard example distribution $\mathcal{D}$ and adversarial example distribution $\mathcal{T}$. Then, given $\lambda \in [0, 1)$ and for any $\epsilon > 0$, with probability at least $1 - \epsilon$,*

$$R_{\mathcal{D}}(f_{AGR}) - R_{S+S'}(f_{AGR}) \leq 2\lambda \hat{\mathfrak{R}}_S(\mathcal{F}) + 3\lambda \sqrt{\frac{\ln(2/\epsilon)}{2N_s}}$$

$$+ (1-\lambda) D_{\mathcal{F}}(\mathcal{D}, \mathcal{T}) + 2(1-\lambda) \hat{\mathfrak{R}}_{S'}(\mathcal{F})$$

$$+ 3(1-\lambda) \sqrt{\frac{\ln(2/\epsilon)}{2N_a}} + \sqrt{\frac{\ln(1/\epsilon)}{2} \left( \frac{\lambda^2}{N_s} + \frac{(1-\lambda)^2}{N_a} \right)}$$

$$\leq 2c\lambda B \frac{(\sqrt{2d \log 2} + 1) \prod_{j=1}^{d} M_F(j)}{\sqrt{N_s}} + 3\lambda \sqrt{\frac{\ln(2/\epsilon)}{2N_s}} \qquad (12)$$

$$+ (1-\lambda) D_{\mathcal{F}}(\mathcal{D}, \mathcal{T}) + 3(1-\lambda) \sqrt{\frac{\ln(2/\epsilon)}{2N_a}}$$

$$+ 2c(1-\lambda) B \frac{(\sqrt{2d \log 2} + 1) \prod_{j=1}^{d} M_F(j)}{\sqrt{N_a}}$$

$$+ \sqrt{\frac{\ln(1/\epsilon)}{2} \left( \frac{\lambda^2}{N_s} + \frac{(1-\lambda)^2}{N_a} \right)}.$$

where $D_{\mathcal{F}}(\cdot, \cdot)$ represents the integral probability metric proposed by [54], $M_F(j)$ denotes the maximum value of the Frobenius norm for each parameter matrix $W_j$, and $d$ is the depth of networks. Similarly, we bound the generalization error in standard adversarial training as:

$$R_{\mathcal{D}}(f_{SA}) - R_{S'}(f_{SA}) \leq 2\hat{\mathfrak{R}}_{S'} + D_{\mathcal{F}}(\mathcal{D}, \mathcal{T}) + 3\sqrt{\frac{\ln(2/\epsilon)}{2N_a}} + \sqrt{\frac{\ln(1/\epsilon)}{2N_a}}$$

$$\leq D_{\mathcal{F}}(\mathcal{D}, \mathcal{T}) + 3\sqrt{\frac{\ln(2/\epsilon)}{2N_a}} + \sqrt{\frac{\ln(1/\epsilon)}{2N_a}}$$

$$+ 2cB \frac{(\sqrt{2d \log 2} + 1) \prod_{j=1}^{d} M_F(j)}{\sqrt{N_a}}.$$

$$(13)$$

In our setting, the ratio of clean to adversarial examples for *AGR* was kept at 1:1 (*i.e.*, $\lambda = 0.5$ and $N_S = N_a$). For conventional adversarial training methods, which usually utilize only the adversarial training dataset, its generalization error can be viewed as Eq. 13, while the error of *AGR* is shown in Eq. 12. It is worth noting that the TRADES method also utilizes the clean examples, and in subsequent experiments, we point out that *AGR* with TRADES as a special case of TRADES is dynamically adjusting the hyperparameters of the loss function of TRADES. Overall, the generalization error of $f_{AGR}$ and $f_{SA}$ are bounded by the empirical training risk, distributed error, and estimation error. The empirical training risk can be optimized to be infinitely small. The distribution error of $f_{SA}$ is two times that of $f_{AGR}$. The remaining term in the inequality with respect to $N_a$ and $N_s$ is denoted the estimation error, which tends to 0 when the training dataset size is infinitely large. Therefore, our *AGR* can achieve a much smaller generalization error. The proof of Theorem 4.3 can be found in Supplementary.

## 5 Experiments

In this section, to verify the effectiveness, efficiency, and feasibility of our proposed *AGR*, we conduct extensive comparative experiments. Firstly, we show that our approach combined with currently available adversarial training methods (*i.e.*, AT [35], TRADES

Haoyu Tong, Xiaoyu Zhang, Yulin Jin, Jian Lou, Kai Wu, and Xiaofeng Chen

**Table 1: The standard accuracy (SA) and robust accuracy of PreAct-ResNet-18 and WideResNet-34 trained on CIFAR10, CIFAR100, and Tiny Imagenet datasets under $l_\infty = 8/255$ against white-box attacks across different defense mechanisms, *i.e.*, AT [35], TRADES [55], MART [45], and AGR (%).**

| Model (Architecture) | Methods | SA | FGSM | PGD-20 | PGD-100 | $CW_\infty$ | AutoAttack |
|---|---|---|---|---|---|---|---|
| CIFAR10 (PreAct-ResNet-18) | AT | $84.86_{\pm0.014}$ | $\mathbf{56.17}_{\pm0.011}$ | $46.30_{\pm0.021}$ | $45.89_{\pm0.008}$ | $46.03_{\pm0.024}$ | $44.01_{\pm0.010}$ |
| | With AGR | $\mathbf{86.22}_{\pm0.004}$ | $55.72_{\pm0.015}$ | $\mathbf{46.85}_{\pm0.026}$ | $\mathbf{46.36}_{\pm0.022}$ | $\mathbf{46.43}_{\pm0.013}$ | $\mathbf{44.91}_{\pm0.009}$ |
| | TRADES | $82.26_{\pm0.165}$ | $57.29_{\pm0.036}$ | $51.12_{\pm0.051}$ | $50.69_{\pm0.042}$ | $50.93_{\pm0.029}$ | $47.06_{\pm0.033}$ |
| | With AGR | $\mathbf{83.30}_{\pm0.012}$ | $\mathbf{58.31}_{\pm0.027}$ | $\mathbf{52.80}_{\pm0.031}$ | $\mathbf{52.54}_{\pm0.033}$ | $\mathbf{52.64}_{\pm0.024}$ | $\mathbf{48.78}_{\pm0.011}$ |
| | MART | $82.33_{\pm0.032}$ | $58.23_{\pm0.005}$ | $51.29_{\pm0.014}$ | $50.82_{\pm0.010}$ | $51.18_{\pm0.021}$ | $46.28_{\pm0.011}$ |
| | With AGR | $\mathbf{83.29}_{\pm0.010}$ | $\mathbf{58.26}_{\pm0.039}$ | $\mathbf{51.64}_{\pm0.060}$ | $\mathbf{51.21}_{\pm0.057}$ | $\mathbf{51.46}_{\pm0.042}$ | $\mathbf{47.34}_{\pm0.034}$ |
| CIFAR10 (WideResNet-34) | AT | $87.03_{\pm0.012}$ | $\mathbf{58.64}_{\pm0.076}$ | $49.06_{\pm0.118}$ | $48.53_{\pm0.063}$ | $48.44_{\pm0.049}$ | $47.71_{\pm0.075}$ |
| | With AGR | $\mathbf{87.53}_{\pm0.007}$ | $58.42_{\pm0.007}$ | $\mathbf{49.51}_{\pm0.009}$ | $\mathbf{49.14}_{\pm0.014}$ | $\mathbf{49.19}_{\pm0.266}$ | $\mathbf{48.36}_{\pm0.018}$ |
| | TRADES | $85.73_{\pm0.012}$ | $58.38_{\pm0.014}$ | $50.39_{\pm0.022}$ | $49.88_{\pm0.020}$ | $50.24_{\pm0.004}$ | $48.41_{\pm0.007}$ |
| | With AGR | $\mathbf{86.50}_{\pm0.023}$ | $\mathbf{60.31}_{\pm0.034}$ | $\mathbf{52.56}_{\pm0.013}$ | $\mathbf{52.04}_{\pm0.011}$ | $\mathbf{52.32}_{\pm0.026}$ | $\mathbf{50.44}_{\pm0.020}$ |
| | MART | $85.14_{\pm0.018}$ | $59.17_{\pm0.058}$ | $50.50_{\pm0.160}$ | $49.96_{\pm0.142}$ | $50.17_{\pm0.074}$ | $47.27_{\pm0.061}$ |
| | With AGR | $\mathbf{86.30}_{\pm0.036}$ | $\mathbf{60.83}_{\pm0.125}$ | $\mathbf{51.37}_{\pm0.350}$ | $\mathbf{50.87}_{\pm0.316}$ | $\mathbf{51.19}_{\pm0.232}$ | $\mathbf{48.74}_{\pm0.117}$ |
| CIFAR100 (PreAct-ResNet-18) | AT | $58.18_{\pm0.048}$ | $27.69_{\pm0.018}$ | $22.04_{\pm0.029}$ | $21.75_{\pm0.024}$ | $21.93_{\pm0.013}$ | $20.19_{\pm0.009}$ |
| | With AGR | $\mathbf{59.18}_{\pm0.016}$ | $\mathbf{27.82}_{\pm0.012}$ | $\mathbf{22.18}_{\pm0.005}$ | $\mathbf{21.78}_{\pm0.008}$ | $\mathbf{22.06}_{\pm0.014}$ | $\mathbf{20.33}_{\pm0.011}$ |
| | TRADES | $53.82_{\pm0.012}$ | $\mathbf{29.84}_{\pm0.014}$ | $\mathbf{27.02}_{\pm0.011}$ | $\mathbf{26.91}_{\pm0.024}$ | $\mathbf{27.13}_{\pm0.021}$ | $\mathbf{23.29}_{\pm0.010}$ |
| | With AGR | $\mathbf{54.53}_{\pm0.009}$ | $29.39_{\pm0.012}$ | $26.85_{\pm0.006}$ | $26.79_{\pm0.014}$ | $26.82_{\pm0.004}$ | $23.27_{\pm0.006}$ |
| | MART | $53.68_{\pm0.015}$ | $29.32_{\pm0.132}$ | $25.35_{\pm0.166}$ | $25.14_{\pm0.154}$ | $25.23_{\pm0.116}$ | $21.67_{\pm0.096}$ |
| | With AGR | $\mathbf{54.21}_{\pm0.105}$ | $\mathbf{29.48}_{\pm0.009}$ | $\mathbf{26.21}_{\pm0.005}$ | $\mathbf{25.95}_{\pm0.032}$ | $\mathbf{26.11}_{\pm0.016}$ | $\mathbf{22.91}_{\pm0.022}$ |
| CIFAR100 (WideResNet-34) | AT | $60.93_{\pm0.079}$ | $31.61_{\pm0.0033}$ | $26.05_{\pm0.008}$ | $25.65_{\pm0.014}$ | $25.88_{\pm0.031}$ | $24.33_{\pm0.023}$ |
| | With AGR | $\mathbf{61.98}_{\pm0.026}$ | $\mathbf{31.68}_{\pm0.027}$ | $\mathbf{26.41}_{\pm0.013}$ | $\mathbf{25.96}_{\pm0.016}$ | $\mathbf{25.99}_{\pm0.007}$ | $\mathbf{24.37}_{\pm0.009}$ |
| | TRADES | $57.10_{\pm0.020}$ | $31.23_{\pm0.029}$ | $26.96_{\pm0.060}$ | $26.75_{\pm0.053}$ | $26.93_{\pm0.046}$ | $24.55_{\pm0.044}$ |
| | With AGR | $\mathbf{58.17}_{\pm0.003}$ | $\mathbf{32.30}_{\pm0.014}$ | $\mathbf{28.34}_{\pm0.012}$ | $\mathbf{28.12}_{\pm0.011}$ | $\mathbf{28.21}_{\pm0.004}$ | $\mathbf{25.84}_{\pm0.016}$ |
| | MART | $57.29_{\pm0.151}$ | $30.33_{\pm0.029}$ | $26.03_{\pm0.062}$ | $25.88_{\pm0.047}$ | $25.96_{\pm0.051}$ | $23.92_{\pm0.044}$ |
| | With AGR | $\mathbf{58.10}_{\pm0.224}$ | $\mathbf{30.52}_{\pm0.019}$ | $\mathbf{27.61}_{\pm0.033}$ | $\mathbf{27.12}_{\pm0.036}$ | $\mathbf{27.24}_{\pm0.025}$ | $\mathbf{25.04}_{\pm0.011}$ |
| Tiny Imagenet (PreAct-ResNet-18) | AT | $31.90_{\pm0.023}$ | $\mathbf{11.09}_{\pm0.016}$ | $\mathbf{8.36}_{\pm0.012}$ | $\mathbf{8.28}_{\pm0.011}$ | $\mathbf{8.12}_{\pm0.008}$ | $6.47_{\pm0.014}$ |
| | With AGR | $\mathbf{35.56}_{\pm0.004}$ | $10.60_{\pm0.014}$ | $7.75_{\pm0.018}$ | $7.76_{\pm0.007}$ | $7.88_{\pm0.008}$ | $\mathbf{6.48}_{\pm0.006}$ |
| | TRADES | $31.06_{\pm0.067}$ | $\mathbf{12.04}_{\pm0.017}$ | $\mathbf{9.99}_{\pm0.011}$ | $\mathbf{10.08}_{\pm0.013}$ | $\mathbf{9.74}_{\pm0.024}$ | $\mathbf{7.20}_{\pm0.019}$ |
| | With AGR | $\mathbf{33.62}_{\pm0.059}$ | $11.47_{\pm0.011}$ | $9.73_{\pm0.009}$ | $9.62_{\pm0.012}$ | $9.65_{\pm0.018}$ | $6.48_{\pm0.010}$ |
| | MART | $29.19_{\pm0.120}$ | $12.65_{\pm0.044}$ | $11.25_{\pm0.012}$ | $10.80_{\pm0.021}$ | $11.03_{\pm0.013}$ | $7.75_{\pm0.018}$ |
| | With AGR | $\mathbf{31.10}_{\pm0.082}$ | $\mathbf{12.94}_{\pm0.012}$ | $\mathbf{11.29}_{\pm0.006}$ | $\mathbf{11.20}_{\pm0.009}$ | $\mathbf{11.06}_{\pm0.015}$ | $\mathbf{8.01}_{\pm0.014}$ |
| Tiny Imagenet (WideResNet-34) | AT | $33.92_{\pm0.032}$ | $10.66_{\pm0.019}$ | $7.57_{\pm0.022}$ | $7.55_{\pm0.023}$ | $7.23_{\pm0.018}$ | $6.54_{\pm0.009}$ |
| | With AGR | $\mathbf{37.30}_{\pm0.032}$ | $\mathbf{11.30}_{\pm0.017}$ | $\mathbf{8.42}_{\pm0.004}$ | $\mathbf{8.30}_{\pm0.007}$ | $\mathbf{8.18}_{\pm0.005}$ | $\mathbf{6.98}_{\pm0.013}$ |
| | TRADES | $32.60_{\pm0.043}$ | $12.22_{\pm0.021}$ | $9.65_{\pm0.034}$ | $9.85_{\pm0.029}$ | $9.47_{\pm0.026}$ | $7.63_{\pm0.013}$ |
| | With AGR | $\mathbf{33.57}_{\pm0.016}$ | $\mathbf{12.89}_{\pm0.008}$ | $\mathbf{11.55}_{\pm0.011}$ | $\mathbf{11.44}_{\pm0.020}$ | $\mathbf{11.25}_{\pm0.015}$ | $\mathbf{8.27}_{\pm0.006}$ |
| | MART | $30.79_{\pm0.136}$ | $13.42_{\pm0.019}$ | $11.82_{\pm0.013}$ | $11.53_{\pm0.025}$ | $11.02_{\pm0.020}$ | $7.37_{\pm0.014}$ |
| | With AGR | $\mathbf{32.96}_{\pm0.093}$ | $\mathbf{13.55}_{\pm0.012}$ | $\mathbf{11.89}_{\pm0.007}$ | $\mathbf{11.65}_{\pm0.023}$ | $\mathbf{11.47}_{\pm0.008}$ | $\mathbf{7.72}_{\pm0.011}$ |

($\lambda = 1/6$) [55], MART [45], AWP [48], Avmixup [25] can significantly improve standard accuracy while ensuring robustness against various attacks. Furthermore, we conducted many ablation studies to confirm the effectiveness of individual components of our methodology. We use the standard accuracy and robustness accuracy as metrics to measure model performance.

## 5.1 Experimental Setup

**Datasets.** For all experiments, we evaluate the standard accuracy and robustness of the proposed *AGR* on CIFAR10 [22], CIFAR-100 [22], and Tiny Imagenet [10], which are three well-known datasets of natural images used in computer vision research. These datasets were divided into two parts, the training set, and the validation set, with a ratio of 5:1. For the data augmentations, we apply 32 × 32 random crops with 4-pixel zero padding, random horizontal flip, and cutout.

**Training.** For CIFAR10/100 and Tiny Imagenet, we utilize PreAct-ResNet-18 [17] and WideResNet-34 [53] architecture as the primary model for evaluation. For the setting of hyperparameters for adversarial training, we train the PreAct-ResNet-18 and WideResNet-34 for 200 epochs by SGD with momentum 0.9, and weight decay of $5 \times 10^{-4}$. The learning rate is initially 0.1, divided by 10 at the 100-th and 150-th calendar times. Regarding the generation of adversarial examples, we use PGD-10 [35] with the value of $\epsilon$ to 8/255, the

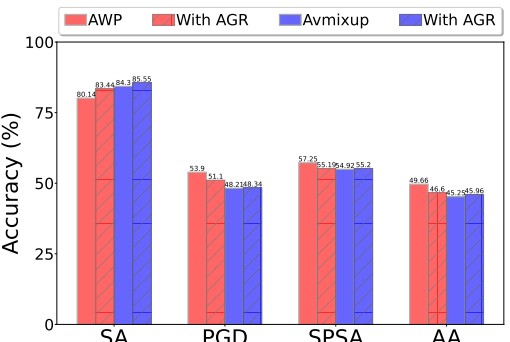

**Figure 3: The standard accuracy (SA) and three robust accuracy of PreAct-ResNet-18 on CIFAR10 across AWP, Avmixup, and *AGR*.**

step size to 2/255, and applied a limit of $l_\infty$ constraint over 10 steps. More detailed settings can be found in the Supplementary.

**Attacks.** For white-box attacks, we consider the four typical attacks below: FGSM [13], PGD-20 [35], PGD-100 [35], and $CW_\infty$ [6]. For black-box attacks, we choose SPSA attack [43] and AutoAttack [9], which contains a black box attack called square attack [4] and three white-box attacks. To ensure consistency in the experimental results, the mean of three experiment repetitions was employed for all results.

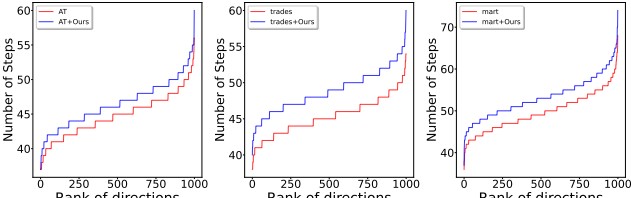

Figure 4: The decision boundary of *AGR*.

Table 2: The robust accuracy of PreAct-ResNet-18 on CIFAR-10/100 and TinyImagenet datasets under under $l_2$ and $l_1$ threat models (%).

| Method | CIFAR10 | | CIFAR100 | | Tiny Imagenet | |
|---|---|---|---|---|---|---|
| | $l_2$ (128/255) | $l_1$ (2000/255) | $l_2$ (128/255) | $l_1$ (2000/255) | $l_2$ (128/255) | $l_1$ (2000/255) |
| AT | $59.73_{\pm 0.018}$ | $47.96_{\pm 0.013}$ | $32.80_{\pm 0.009}$ | $30.26_{\pm 0.012}$ | $\mathbf{14.40}_{\pm 0.015}$ | $26.31_{\pm 0.014}$ |
| Ours | $\mathbf{60.76}_{\pm 0.013}$ | $\mathbf{48.84}_{\pm 0.011}$ | $\mathbf{33.81}_{\pm 0.016}$ | $\mathbf{31.09}_{\pm 0.012}$ | $14.31_{\pm 0.008}$ | $\mathbf{26.52}_{\pm 0.010}$ |
| TRADES | $61.89_{\pm 0.042}$ | $45.82_{\pm 0.036}$ | $34.88_{\pm 0.013}$ | $32.68_{\pm 0.017}$ | $16.32_{\pm 0.020}$ | $28.35_{\pm 0.031}$ |
| Ours | $\mathbf{63.14}_{\pm 0.035}$ | $\mathbf{47.54}_{\pm 0.037}$ | $\mathbf{35.20}_{\pm 0.028}$ | $\mathbf{33.94}_{\pm 0.013}$ | $\mathbf{17.38}_{\pm 0.017}$ | $\mathbf{29.55}_{\pm 0.025}$ |
| MART | $61.65_{\pm 0.011}$ | $48.01_{\pm 0.017}$ | $34.37_{\pm 0.015}$ | $31.43_{\pm 0.013}$ | $16.17_{\pm 0.009}$ | $27.42_{\pm 0.010}$ |
| Ours | $\mathbf{61.94}_{\pm 0.049}$ | $\mathbf{48.11}_{\pm 0.035}$ | $\mathbf{34.50}_{\pm 0.013}$ | $\mathbf{31.97}_{\pm 0.007}$ | $\mathbf{17.55}_{\pm 0.009}$ | $\mathbf{27.87}_{\pm 0.011}$ |

## 5.2 Main Results

**Impact on standard accuracy.** Table 1 shows that our proposed *AGR* method, when combined with diverse adversarial training methods, can significantly enhance the standard accuracy of the model on CIFAR10, CIFAR100, and Tiny Imagenet. Specifically, when applied in conjunction with adaptive gradients reconstruction, *AGR* achieves an impressive standard accuracy of 86.22% on CIFAR10 with PreAct-ResNet-18. This is a significant improvement compared to the baseline AT, which only manages to achieve 84.86% accuracy on clean images, resulting in a gap of 1.36%. Despite the gap in standard accuracy, the robustness accuracy of both methods is quite comparable, with only a marginal difference of 0.49%. The identical experimental results can also be observed on the Tiny Imagenet dataset.

**Robustness against white-box attacks.** To verify the reliability of our approach, we conducted a thorough evaluation of its robustness against a range of white-box attacks. Specifically, we considered various attacks with the same norm constraints (e.g., $l_\infty = 8/255$). The results, as presented in Table 1, demonstrate that our proposed method, which incorporates adaptive gradient reconstruction, consistently maintains exceptional robustness across all evaluated attacks. Even in the worst-case scenario, where the model was trained with TRADES on CIFAR100, there was only a 0.45% reduction in robustness. Over the Tiny Imagenet dataset, we also demonstrate the effectiveness of our method, which performs well in improving generalization. In addition to this, we further consider other adversarial training methods in conjunction with *AGR*, such as AWP [48], Avmixup [25]. The results are shown in Figure 3, where the proposed method is effective in improving generalization while maintaining robustness.

To demonstrate the robustness of our method against different white-box adversarial attacks, we evaluate our method against FGSM, PGD-100, and C&W. As shown in Table 1, our method is efficient in maintaining robustness against diverse adversarial attacks. We can observe that the combination of our *AGR* method also maintains reasonable robustness against unseen perturbations.

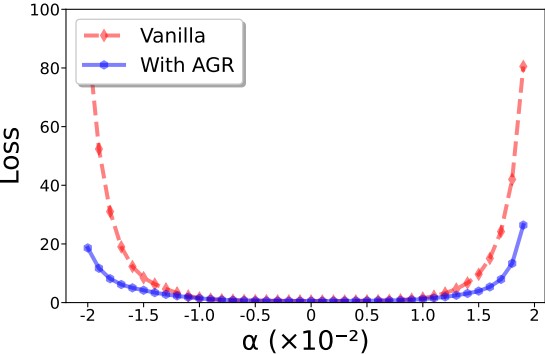

Figure 5: Comparison of the weight loss landscape for vanilla AT and AGR of PreAct-ResNet-18 trained on CIFAR10. These curves are the change in loss when moving model weight in the direction of a randomly sampled from a Gaussian distribution with the step size of $\alpha$.

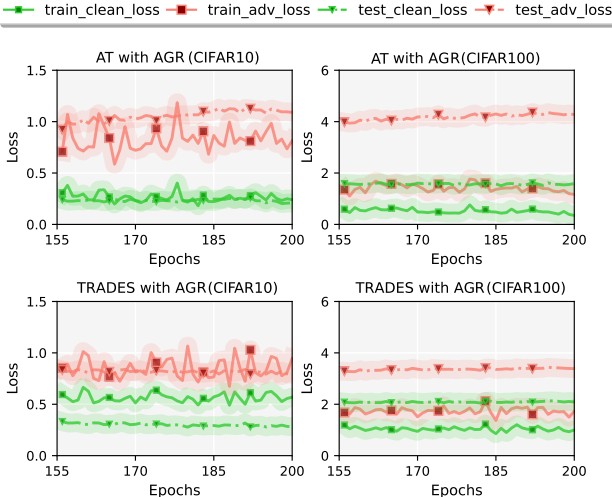

Figure 6: The loss change with respect to epochs of AT-AGR and TRADES-AGR of PreAct-ResNet-18 trained on CIFAR10 and CIFAR100.

**Robustness against black-box attacks.** We also conducted two black-box attack tests on our approach, *i.e.*, query-based attack SPSA and AutoAttack, assessing robustness by a black-box attack (Square Attack) and three white-box attacks. Table 1 and Figure 3 illustrate our approach's effectiveness in defending against black-box attacks.

**Visualization results.** To visualize that our method improves generalization, we show the decision boundary of *AGR* in Figure 4 and the weight loss landscape of the model trained with vanilla AT and *AGR* in Figure 5. For decision boundary, we randomly select an image and generate 1K random directions, applying continuously perturbations with a fixed step size ($\alpha = 1/255$) along each direction until the model's prediction for the image changes. We record the number of steps required to change the prediction result of the example in 1K directions in Figure 4 in ascending order. The decision boundary of our method is further away from the example than the baseline. Weight loss landscape is a commonly used measure to describe the generalization gap in standard training [28, 33]. We can observe that our approach has a much flatter loss compared to

**Table 3: Comparison of DNNs model trained with/without proposed GOP in different adversarial training methods trained on CIFAR10 (%).**

| Method | Standard acc | Robust acc | GOP frequency |
|---|---|---|---|
| Vanilla AT | $86.14_{\pm0.011}$ | $46.42_{\pm0.018}$ | - |
| + GOP | $\mathbf{86.22}_{\pm0.004}$ | $\mathbf{46.85}_{\pm0.026}$ | 2.03 |
| TRADES | $82.98_{\pm0.014}$ | $52.60_{\pm0.024}$ | - |
| + GOP | $\mathbf{83.30}_{\pm0.012}$ | $\mathbf{52.80}_{\pm0.031}$ | 14.93 |
| MART | $82.94_{\pm0.008}$ | $51.43_{\pm0.032}$ | - |
| + GOP | $\mathbf{83.29}_{\pm0.010}$ | $\mathbf{51.64}_{\pm0.060}$ | 6.09 |

**Table 4: Comparison of the effects of three interpolation methods on the generalization and robustness of vanilla AT trained on CIFAR10. Case 1 is to average the gradients of standard and robust loss, Case 2 is to mix the two gradients in a certain proportion(*i.e.*, $\tau = 0.9$), while Case 3 is to apply GICS (%).**

| Method | AT | | TRADES | | MART | |
|---|---|---|---|---|---|---|
| | Clean | PGD | Clean | PGD | Clean | PGD |
| + Case 1 | $86.04_{\pm0.005}$ | $46.70_{\pm0.013}$ | $\mathbf{83.32}_{\pm0.011}$ | $52.64_{\pm0.017}$ | $83.28_{\pm0.008}$ | $51.55_{\pm0.015}$ |
| + Case 2 | $85.10_{\pm0.010}$ | $48.42_{\pm0.019}$ | $82.48_{\pm0.012}$ | $52.84_{\pm0.017}$ | $82.54_{\pm0.011}$ | $52.55_{\pm0.022}$ |
| + Case 3 (Ours) | $\mathbf{86.22}_{\pm0.004}$ | $46.85_{\pm0.026}$ | $83.30_{\pm0.012}$ | $52.80_{\pm0.031}$ | $\mathbf{83.29}_{\pm0.010}$ | $51.64_{\pm0.060}$ |

vanilla AT, which indicates better generalization. As depicted in Figure 6, we show that our method achieves a remarkably smooth loss trained by vanilla AT and TRADES on CIFAR-10/100 datasets. The loss change result on Tiny Imagenet can be found in Supplementary. **Robustness against unseen attacks.** We have conducted evaluations for other threat models. Table 2 reports the adversarial robustness using PRN-18 under $l_2$ and $l_1$ threat models, which indicates that the proposed method is also effective for other threat models.

## 5.3 Ablation Study

We aim to provide a thorough analysis of the proposed *AGR* method by conducting substantial ablation studies on each individual component. Our primary objective is to evaluate the effectiveness of the Gradient Orthogonal Projection (GOP) and Gradient Interpolation Based on Cosine Similarity (GICS) of *AGR* for standard accuracy enhancement, respectively.

**Evaluation on gradient orthogonal projection.** To evaluate GOP's ability to improve standard accuracy, we tested models trained with and without GOP, comparing it to vanilla AT, TRADES, and MART under consistent experimental settings. The results, as shown in Table 3, indicate that GOP significantly enhances standard accuracy in all adversarial training. Notably, TRADES and MART with GOP show greater improvement than vanilla AT, likely due to the frequency of orthogonal projection during training. In AT-GOP, GOP frequency is 2.03%, while in TRADES and MART, it reaches 14.93% and 6.09%, respectively.

**Evaluation on gradient interpolation.** We conducted a comparative experiment with three gradient interpolation methods: average gradients, fixed scale interpolation, and our GICS (Section 3.4). Table 4 shows that our method significantly improves standard accuracy, but in some cases, it results in decreased robustness compared to Case 2.

In particular, we point out that the interpolation method in TRADES is actually changing the hyperparameters $\lambda$ of the loss function of TRADES, whereas our method (*i.e.*, Case 3) dynamically

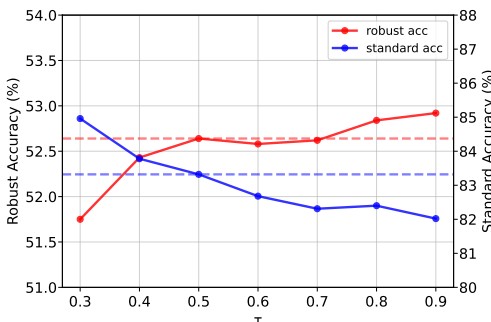

**Figure 7: Comparison of robust and standard accuracy for TRADES-GI of PreAct-ResNet-18 on CIFAR10 under different $\tau$ values in Case 2. The red and blue dashed lines indicate TRADES-AGR results in Case 3.**

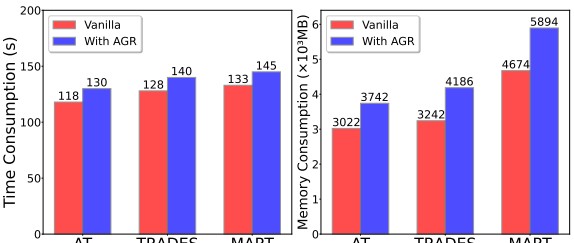

**Figure 8: Comparison of time and memory consumption for PreActResNet-18 trained with AT, TRADES, MART, and *AGR* on CIFAR10. (Left) shows seconds per epoch, (Right) shows memory usage by each method.**

adjusts $\lambda$ so that when robustness is lost too much, interpolation yields a larger $\lambda$ to ensure that robustness is not lost. In this way, we can get a better balance between standard accuracy and robustness accuracy compared to fixed interpolation weights. The results of Case 2 with different $\tau$ values are shown in Figure 7. As $\tau$ increases, robust accuracy increases while standard accuracy decreases, indicating a static trade-off where the gradient focuses more on robustness over generalization. In Case 3, this trade-off is dynamically managed by adjusting the interpolation weight based on the gradient directions of clean and adversarial losses. As shown by the dotted line in Figure 7, Case 3 achieves a better balance between robustness and generalization compared to Case 2, often providing higher generalization with comparable robustness.

**Evaluation on time and memory consumption.** Here we demonstrated the time and GPU memory consumption of our proposed method compared to the vanilla AT method in Figure 8. Time loss and memory usage are still increased compared to the original method.

## 6 Conclusion

In this paper, we perform a comprehensive study on how to balance generalization and robustness in adversarial training. From a novel perspective, we proposed Adversarial Training with Adaptive Gradients Reconstruction (*AGR*) to implement the gradient information of clean and adversarial examples to dynamically handle the trade-off between generalization and robustness in order to improve standard accuracy. Extensive experiments show that our method has an excellent performance in improving generalization while maintaining robustness.

## Acknowledgments

This work is supported by the National Natural Science Foundation of China (No. 62102300, 62206207, 61960206014, and 62121001), Guangdong Provincial Key Laboratory of Novel Security Intelligence Technologies (No. 2022B1212010005), Fundamental Research Funds for the Central Universities (No. ZYTS24140), and '111 Center' (No. B16037).

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
