# OpenReview forum: "Balancing Generalization and Robustness in Adversarial Training via Steering through Clean and Adversarial Gradient Directions"
_acmmm.org/ACMMM/2024/Conference — MM2024 Poster_

### Official Review · Reviewer_4R6b · 2024-04-29

**Rating:** 5
**Confidence:** 3

**Summary:**

The authors introduce AGR, a novel method aimed at balancing generalization and robustness in adversarial training, and evaluate its effectiveness across various benchmark methods and datasets.

**Strengths:**

1. The perspective is novel and interesting. The author explores the gradient space between standard and adversarial examples and uses GOP and GICS to adjust negative and positive correlations.

2. The proposed methodology stands out as technically sound and effective both theoretically and empirically.

3. The experimental results convincingly validate the approach.

4. The paper is overall well-written and easy to follow.

**Limitations:**

While the current version of the paper is satisfactory, addressing the following points would still be meaningful.

1. Figure 1 illustrates that the ratio of positive to negative gradients varies by layer. Conducting a layer-dependent ablation study could yield insightful findings about how different layers contribute to the effectiveness of AGR.

2. The results in Figure 3 indicate that incorporating AGR results in diminished robustness with AWP. Could the author provide an intuitive explanation for this phenomenon?

3. It would be beneficial to test on adaptive attacks, such as BPDA and Lagrange.

4. This paper does not involve any multimedia or multimodal elements.

**Suitability:**

1

---

### Official Review · Reviewer_BK4D · 2024-05-15

**Rating:** 4
**Confidence:** 2

**Summary:**

This paper presents a comprehensive study on how to balance generalization and robustness in adversarial training. They first propose adversarial training with adaptive gradient reconstruction (AGR), which utilizes the gradient information of both clean and adversarial examples to dynamically handle the trade-off between generalization and robustness, thus improving the criterion accuracy. Since I am not very familiar with the field, I will be adjusting my scores based on the ratings of other reviewers.

**Strengths:**

1.	The article proposes a very new perspective to improve the generalization ability and performance of the model through gradient orthogonal projection. I think it is very innovative.
2.	The experimental part of the article is very detailed and argues very favorably for the validity of the model.

**Limitations:**

1.	The article suggests that the performance of testing clean data is degraded under robust modeling, and I would like to know what kind of data the authors hope to be able to achieve excellent performance with, and how it can be demonstrated in the experiments?
2.	What is the strong connection between articles and multimedia?
3.	What was the original purpose of introducing the gradient space to improve the model's ability to generalize to adversarial data? How does the experimental component demonstrate the model's ability to generalize?

**Suitability:**

2

---

### Official Review · Reviewer_tD1w · 2024-05-22

**Rating:** 4
**Confidence:** 4

**Summary:**

To trade-off between clean and robust accuracy from the gradient space, this paper proposes Adversarial Training with Adaptive Gradient Reconstruction (AGR). This method optimizes adversarial gradient direction and interpolation scheme to increase generalization without compromising robustness. Besides, theoretical analysis and empirical experiments demonstrate that AGR effectively balances generalization and robustness, achieving better performance.

**Strengths:**

For presentation, the paper is well organized and interesting. The proposed method is easy to follow, which have visual description. For theoretical contribution, the author proposed and analyzed a theorem on the generalization properties of AGD. Third, this paper proposes a method from the perspective of gradients that can enhance the clean and robust accuracy of models, which may inspire progress in other related works. A detailed evaluations are conducted; the results show the proposed method enhances the performance of existing adversarial training as a plug-in.

**Limitations:**

1.	Some definitions are not consistent, such as “example” and “sample,” which leads confusion during reading.
2.	The improvement in robust accuracy is limited, particularly when against attack in larger datasets and AutoAttack.
3.	Considering that the proposed method requires computing gradients with respect to both clean and adversarial examples, it may require increased computational costs. Therefore, it is necessary to conduct comparative experiments or analyses regarding training time to determine the competitiveness in efficiency. Additionally, the authors can also provide constructive suggestions to relieve this issue.
4.	I recommend authors include pseudocode descriptions of the algorithm and link them to the corresponding formulas to assist readers in better understanding.

**Suitability:**

2

---

### Meta-Review · Area_Chair_DLLS · 2024-07-01

**Recommendation:** Accept (Poster)
**Confidence:** 5

**Metareview:**

The paper received three positive reviews, but reviewers mainly focus on its relevance to the multimedia field. The authors provided a detailed explanation of this issue in their rebuttal. Therefore, I am inclined to accept the paper. However, upon acceptance, the authors should emphasize the paper’s relevance to the multimedia field.